# Hepatitis B Vaccination Coverage among Bangladeshi Healthcare Workers: Findings from Tertiary Care Hospitals

**DOI:** 10.3390/vaccines11010041

**Published:** 2022-12-24

**Authors:** Md. Golam Dostogir Harun, Shariful Amin Sumon, Tahrima Mohsin Mohona, Aninda Rahman, Syed Abul Hassan Md Abdullah, Md. Saiful Islam, Md. Mahabub Ul Anwar

**Affiliations:** 1Infectious Diseases Division, International Centre for Diarrhoeal Disease Research, Bangladesh (ICDDR, B), Dhaka 1212, Bangladesh; 2Communicable Disease Control (CDC), Directorate General of Health Services, Government of Bangladesh, Dhaka 1212, Bangladesh; 3SafetyNet Bangladesh, Dhaka 1212, Bangladesh; 4National Centre for Epidemiology and Population Health, Australian National University, Canberra 2601, Australia; 5Office of Health Affairs, West Virginia University, Morgantown, WV 26506, USA

**Keywords:** vaccination, hepatitis B, healthcare workers, factors, Bangladesh

## Abstract

Healthcare workers (HCWs) are at a four-fold higher risk of being infected with the hepatitis B virus in hospital settings. This study investigated the hepatitis B vaccination coverage among Bangladeshi HCWs in selected tertiary care hospitals. Between September 2020 to January 2021, a multicenter cross-sectional study was conducted in 11 hospitals across Bangladesh. Participants included physicians, nurses, cleaners, and administrative staff. A semi-structured questionnaire was used to collect data through face-to-face interviews. Descriptive and multivariate statistics were used to analyze the data. The overall hepatitis B vaccination coverage was 66.6% (1363/2046) among HCWs, with cleaning staff having the lowest at 38.8%. Among the unvaccinated, 89.2% of HCWs desired to receive the free vaccine in the near future. In the last year, over one-fourth of staff (27.9%) had at least one history of needlestick injury. Only 9.8% HCWs were found to have attended training on hepatitis B virus prevention and management in the previous two years. Multivariate analysis revealed that physicians (AOR: 7.13, 95% CI: 4.94–10.30) and nurses (AOR: 6.00, 95% CI: 4.09–8.81) were more likely to be vaccinated against hepatitis B than cleaners and administrative staff. Low uptake of hepatitis B vaccination among HCWs suggests policies that require vaccination are needed to achieve optimum vaccine coverage.

## 1. Introduction

Hepatitis B is a highly infectious disease that is easily transmitted through parenteral contact with infected blood and other bodily fluids [1,2]. It is a major public health threat that affected 296 million people around the world until the year 2019 [3]. Healthcare workers (HCWs) are at high risk of hepatitis B virus (HBV) infection due to frequent exposure to bloodborne pathogens. Occupational exposure to HBV can occur through accidental needlestick injuries (NSIs) while caring for HBV-positive patients or from contaminated hospital surfaces, as the virus can remain viable for up to 7 days outside the body [4,5]. The World Health Organization (WHO) estimated 2 million HCWs sustain needlestick injuries annually, from which 3.3% develop hepatitis B following sharps injury [6]. HCWs all over the globe are therefore classified as a high priority by international health organizations and are recommended to take the hepatitis B vaccine as a standard precaution [7,8].

According to the WHO, hepatitis B is responsible for 3.8% of global disease burden, while 820,000 deaths were estimated to occur due to this infection [9]. In Bangladesh, the prevalence of hepatitis B was 4.0% among the general population and as high as 8% among HCWs [10,11]. The effects of hepatitis B virus (HBV) include liver cancer, cirrhosis, and other conditions caused by chronic viral hepatitis [1,12]. An affordable and safe vaccine to protect against HBV infection has been available since 1981, and when three doses of vaccines are completed, it is deemed the most effective method of protection against hepatitis B [13]. Due to the goal set by WHO’s World Health Assembly (WHA) to eliminate viral hepatitis by 2030, global efforts have been made to improve vaccine coverage, especially among HCWs. Yet, there is a substantial disparity in vaccination coverage globally; immunization coverage varies from 90% in the western Pacific and the Americas to 75% in African regions [14,15]. Studies stated that approximately two million HCWs are at risk of occupational exposure to HBV every year [16]. The Centers for Disease Control and Prevention (CDC) deem HCWs as a population at high risk for HBV infection and recommend vaccination against it [5,8], yet no recommendation or policy for HCWs exists in Bangladesh. The Government of Bangladesh has since 2003 included the hepatitis B vaccine in the Expanded Programme on Immunization (EPI) free of cost, but this benefit has not been extended to adults [17]. Factors such as the perceived threat and severity of hepatitis B infection, awareness regarding the availability of hepatitis B vaccines, cost of the vaccine, and the cadre of HCWs influence the vaccine uptake [18,19,20].

Despite the high incidence of HBV infection among HCWs in the country, the hepatitis B vaccination status in this professional group has not been studied yet. At a moment when a shortage of healthcare workforce has been reported [14], such adverse consequences for HCWs will reduce the availability of health services and further weaken the health system. Therefore, the purpose of this study was to determine the coverage of hepatitis B vaccination, willingness to receive the vaccine, and factors influencing hepatitis B immunization among HCWs of selected tertiary care hospitals in Bangladesh. The findings would help in identifying and generating of evidence to support policymakers in developing tailored campaigns for increasing the vaccine uptake among HCWs and preventing vaccine-preventable disease outbreaks in hospital settings.

## 2. Materials and Methods

### 2.1. Study Design and Settings

A multicentered, cross-sectional study was carried out from September 2020 to January 2021. We conducted this survey across 11 tertiary care hospitals across Bangladesh, including nine government and two private hospitals. These 11 hospitals were purposively selected by the government authorities (the Ministry of Health) and represent a quarter of all the country’s tertiary level hospitals, and a third of the public facilities of this scale. Tertiary, the highest level, hospitals or institutes have medical colleges and universities, specialized departments, and are usually referral hospitals, where patients from all over the country seek medical help. The average bed capacity and annual patient turnover of selected hospitals ranged from 450–2600 and 15,000–85,000, respectively. The high patient volume makes the transmission of infectious diseases a serious concern in these facilities.

### 2.2. Participants

We enrolled both clinical and non-clinical staff in this study. The participants included physicians, nurses, cleaning staff, and administrative staff from the departments of Medicine, Pediatrics, Surgery, and Gynecology and Obstetrics. Before initiating the study, we obtained written permission from the Directorate General of Health Services (DGHS), the division under the Ministry of Health and Family Welfare responsible for health services in Bangladesh, and also circulated the official letter to each hospital. After seeking permission from the respective hospital director/administrator and department heads, we approached the HCWs for an interview. Data were collected through face-to-face interviews with participants in the local (Bengali) language. HCWs who were on duty during data collection and agreed to participate were eligible for this survey. Approximately 25% of HCWs of each category were interviewed randomly from each study hospital.

### 2.3. Data Collection

A pre-tested semi-structured questionnaire composed of two main sections was developed to collect data. The first section comprised socio-demographic characteristics such as age, sex, educational attainment, occupational experience, profession type, and working department. The other segment included hepatitis B vaccination status, willingness to be vaccinated, history of needlestick injuries (NSIs). We also asked the respondent whether he/she had ever carried out a blood test of HBsAg to assess the presence of the hepatitis B virus, history of attending training in hepatitis B prevention, diagnosis, or management, and vaccination required for HCWs. HCWs who had received all three doses of hepatitis B vaccination were considered vaccinated for this survey. The responses for vaccination status, blood test history against the hepatitis B virus, or the number of needlestick injuries were determined based on the respondent’s recall. Before administering the face-to-face interview, participants were informed about the study’s objectives and assured of strict confidentiality, privacy, and anonymity of collected information. Furthermore, we informed them that participation in this study was entirely voluntary and they have full rights to opt out without any consequence.

### 2.4. Statistical Analyses

We performed statistical analysis using STATA version 13. Descriptive and multivariate statistics were used to analyze the data. The mean with standard deviation (SD) and frequencies with percentages were used to summarize continuous and categorical variables. Responses to the open-ended question of “Number of needlestick injuries” and “Recommended vaccines that are essential for HCWs” were identified and categorized. Considering the cluster effect of different hospitals, mixed effects logistic regression analysis was conducted to assess the associated factors of hepatitis B vaccination uptake with HCW type, age group (18–28, 29–39, 40–50, and >50 years), sex, working experience, and vaccine recommendation. Variables with a *p*-value of 0.25 from the univariate model were included in the multivariate analysis. Additionally, the multicollinearity of independent variables was verified before counting them for the model. We present both unadjusted odds ratio (UOR) and adjusted odds ratio (AOR) data with a 95% confidence interval (CI). A *p*-value of less than 0.05 was regarded as statistically significant throughout the study.

### 2.5. Quality Control

The study design, data collection guidelines and procedures, quality checking plan, and data management, including analysis, were all guided by research specialists. Enrolled data collectors were experienced in data collection from health facilities, and we performed pre-tests of the questionnaire after adequate hands-on training sessions. During the interview, sufficient time was given to the respondents to recall the information. We conducted multilevel supervision and spot-checking of the questionnaire to ensure the quality of the collected data.

## 3. Results

### 3.1. General Characteristics of Participating HCWs

A total of 2,046 HCWs participated in this study with a 96% response rate. The key demographic and professional characteristics of the participants are presented in Table 1. Nurses (934, 45.7%) were the most enrolled HCWs, followed by physicians (526, 25.7%). Most of the participants (86.9%) were aged 18–39 years, with 62.3% of HCWs being female. Less than half (45.3%) of HCWs had five years or less of work experience. All physicians and nurses held specialized degrees (Bachelor of Medicine, Bachelor of Surgery (MBBS) or bachelor’s degree, or Diploma in Nursing). However, 45.7% of the administrative staff were bachelor’s degree or diploma holders, while the majority (95.1%) of the cleaning staff had a secondary degree or lower. Around one-third (30.8%) of participants were enrolled in the Medicine department. Regarding hospital ownership, the vast majority (87.7%) of HCWs were working in public hospitals in this study.

### 3.2. Hepatitis B Vaccination Coverage among Participating HCWs

The vaccination coverage rates are displayed in Table 2. Overall, 66.6% (95% CI 64.6–68.7) of HCWs were reported to have received the hepatitis B vaccine. More than three-quarters of physicians and nurses obtained the hepatitis B vaccine, but only 41.1% of administrative staff and 38.8% of cleaning staff were found to be vaccinated against hepatitis B. Nearly two-thirds (65.0%) of participants had a HBsAg blood test to determine the presence of hepatitis B infections. Only 53.5% of HCWs had performed both a blood test and received hepatitis B vaccine. Most unvaccinated respondents (614/683, 89.2%) desired to receive free vaccines in the future. Only 9.8% of HCWs attended training on hepatitis B virus prevention, diagnosis, or management in the last two years. In the past year, over one-fourth (27.9%) of HCWs reported a history of NSI, with a mean of 2.2 ± 0.8 times. Among the HCWs with a history of NSI, 26.3% were unvaccinated against hepatitis B. The majority of respondents (67.0%) reported taking extra precautions when in contact with chronic hepatitis B-positive patients. However, 62.3% of respondents agreed the hepatitis B vaccine was essential and should be made mandatory for HCWs in our country.

### 3.3. Factors Associated with Hepatitis B Vaccination Coverage

Table 3 depicts the findings of logistic regression analyses, including vaccine uptake status by category. Most vaccinated HCWs were nurses and physicians (53.2% and 29.5%), while administrative and cleaning staff accounted for only 9.7% and 7.6%, respectively. More than 40% of HCWs between the ages of 29 and 39 received the vaccine. However, the rate of vaccination among workers with five years or less of experience was 42.3%. Among vaccine recipients, 65.6% recommend the hepatitis B vaccine as essential for HCWs in our country. In the multivariable model, the association remained between hepatitis B vaccine uptake and the participant’s category, age, sex, working experience, and vaccine recommendations. Physicians (AOR: 6.83, 95% CI: 4.72 -9.89) and nurses (AOR: 6.02, 95% CI: 4.19–8.65) were more likely to get vaccinated against hepatitis B compared to administrative staff. The likelihood of having the hepatitis B vaccine was decreased among older HCWs. Sex did not affect vaccination status and was not associated with hepatitis B vaccination coverage rates. The odds ratio of receiving the hepatitis B vaccine among female HCWs was 1.06 (95% CI: 0.81–1.40) compared to males, which was statistically significant in the UOR. When compared to HCWs who had been working for more than ten years, newly employed (≤5 years of working experience) HCWs were 36% more likely to have a non-vaccination status (AOR: 0.64, 95% CI: 0.43–0.95, *p*-value 0.026).

## 4. Discussion

This multicenter study investigated the coverage of the hepatitis B vaccine, including its associated factors among HCWs in selected Bangladeshi tertiary care hospitals. We found that, overall, two-thirds of HCWs had hepatitis B vaccination coverage at tertiary level health facilities. This finding is consistent with similar studies in different low- and middle-income countries (LMICs) that reported hepatitis B vaccine coverage ranging from 56.9% to 69.1% [19,21,22]. Amongst vaccinated HCWs, physicians and nurses were more likely to receive the hepatitis vaccine [23]. Findings indicate that the cleaning staff had the lowest vaccine coverage, which is comparable with several previous studies that documented inadequate hepatitis B vaccination for cleaning staff [24,25,26]. One plausible explanation for this negligible uptake could be that cleaners are not prioritized in terms of getting training and vaccination. This subpar vaccination coverage among cleaners is a particular concern as cleaners in resource-constrained settings are frequently burdened with tasks unrelated to their primary role [27]. This implies the requirement for a national policy level implementation of mandatory vaccination to increase hepatitis B vaccination coverage among HCWs.

Most (89.2%) unvaccinated HCWs in the survey expressed willingness to receive the hepatitis B vaccine in the future, if available and free of charge. According to a Chinese study published in 2019, HCWs who receive free vaccination at their workplace were 1.4 times more likely to get vaccinated than HCWs who did not receive the free vaccine [22]. Another study in Cameroon found that HCWs with a history of free hepatitis B screening were nearly 30 times more likely to be fully vaccinated [25]. One of the major barriers to vaccination coverage is the high cost of the hepatitis B vaccine. A study from Ethiopia cited that about 45% of HCWs had not received their vaccinations, considering the high price of hepatitis vaccines [28]. This indicates that the Ministry of Health in Bangladesh should implement free and mandatory vaccination programs to increase vaccination uptake among HCWs in hospital settings. Additionally, wide dissemination and regular campaigns were needed to raise awareness of and encourage all HCWs to receive hepatitis B immunization.

The present study documented high numbers of occupational needlestick injuries (NSIs) among HCWs. Over one-fourth (27.9%) of HCWs had at least one history of NSI during the last 12 months, with nursing staff having the highest incidence. These findings are in line with the published results from China (23.2%) and Saudi Arabia, where a quarter (24.9%) of HCWs reported the incidence of NSI [29,30]. However, other LMICs, such as Iran (42.5%) and Nigeria (51.0%), revealed higher NSI history among HCWs [31,32]. Furthermore, we found that one-tenth of HCWs with a history of NSI had not yet received hepatitis B vaccination, putting them at risk of infection. The implications of HBV infection are well documented as one of the toughest bloodborne occupational hazards among HCWs [33], and hepatitis B vaccination is considered the most effective and viable system to protect both HCWs and patients from this life-threatening disease [34]. Regardless of HCW status, tailored and regular vaccination programs can reduce the risk of occupational NSI in hospitals.

Findings from the study indicate that a meager percentage of HCWs participated in training on hepatitis B virus prevention, diagnosis, or management in the last two years, particularly the cleaning and administrative staff. Studies conducted on HCWs in Bangladesh and India show that cleaning staff are often not prioritized in terms of training and are either completely excluded or given ‘’informal’’ orientation training on infection prevention or they ‘’learn on the job’’ [35,36,37]. Although trained cleaning staff proved to be an asset in reducing HAIs in a similar resource-constrained setting [38], less than 33% of facilities surveyed across Bangladesh, India, Gambia, and Zanzibar delivered formal training to their cleaning staff [27]. In this study, approximately one-tenth of HCWs expressed no interest in receiving the hepatitis B vaccine in the future. In addition, we observed that almost all cleaning staff’s educational attainment was secondary or lower, which makes them less knowledgeable about hepatitis B virus prevention, including management. Our findings are supported by previous reviews and studies, which revealed that cleaners with an education level of less than a diploma were 53% less likely to get full-dose vaccination of hepatitis B than those with an education level above a diploma [39]. This phenomenon is not only limited to developing countries as a US study also documented that higher education level was also associated with greater hepatitis B vaccination coverage [40]. Therefore, it is imperative to arrange periodic refresher training, while ensuring the participation of all cadres of hospital staff, to raise awareness and knowledge of the factors influencing hepatitis B vaccine uptake in hospital settings.

From the multivariate logistic analysis, working experience of ≥5 years was allied with an increased likelihood of vaccination uptake (AOR: 0.64, 95% CI 0.43–0.95). These findings contrast with a previous study which revealed HCWs with more than ≥ 10 years of work experience were 2.3-fold more likely to get the hepatitis B vaccine [25]. Another Ethiopian study documented that the odds of hepatitis B vaccination increased by 7.27-fold as the work experience increased in HCWs [41]. Likewise, other studies in similar settings found that HCWs with work experience of up to 5 years were 66% less likely to get full-dose hepatitis vaccine than those with five years or more. Although the gender distribution of the investigated sample of physicians and nurses in our study is comparable to a national report, overall, the HCWs were found to consist of a greater proportion of females. This may be because our study also investigated cleaning staff as part of the HCW cohort, who were not accounted for in the national survey. From the univariate model, our analysis revealed that female HCWs were more likely to get vaccinated for hepatitis B compared to their male counterparts (AOR: 2.02, 95% CI 1.65–2.48, *p*-value 0.000). This agreed with a recent systematic review and meta-analysis conducted in Ethiopia, which documented that male HCWs were 35% less likely to get the full-dose hepatitis B vaccine than their female counterparts [39]. A possible explanation for this might be that women have a higher risk perception and might be more concerned about their health compared to men.

Further research in Bangladesh concerning hepatitis B infection and vaccination in hospitals and among healthcare workers is required. Implementing the national policy for mandatory hepatitis B vaccination, along with regular training and awareness regarding hepatitis B prevention and treatment, is crucial to enhance optimal vaccine coverage among HCWs in Bangladesh. Hospital administrators and leadership teams can play a critical role in ensuring full vaccination coverage for all HCWs, including newly joined and outsourced staff working for their hospital.

A few limitations of our study included participants’ recall bias in remembering vaccination status, number of NSIs, and the amount of training received, inability to report the vaccine dosages through a vaccine card, and inability to explain the causal association due to the cross-sectional study design, which might have caused over- or underreporting. However, adult vaccinations and NSIs are events that HCWs are less likely to forget; therefore, recall bias has not affected our findings.

## 5. Conclusions

The overall hepatitis B vaccination coverage among Bangladeshi HCWs was found to be low, giving insight for hospital leadership and policymakers to work on this issue. The HBV vaccination should be endorsed in the hospital vaccination policy, aligned with the national policy. All levels of HCWs should be educated about the necessity of vaccination against HBV, and vaccination strategies and systems should be in place in the hospital. In addition, vaccination policies are required for all permanent, temporary, and outsourced healthcare workers to be vaccinated immediately after joining the hospital.

## Figures and Tables

**Table 1 vaccines-11-00041-t001:** General characteristics of participating healthcare workers (HCWs).

Variables	Physicians (N = 526)	Nurses (N = 934)	Cleaning Staff (N = 268)	Admin Staff (N = 318)	Total (N = 2046)
			n (%)		
**Age (in years)**					
Mean (SD)	31.2 (±7.2)	33.4 (±8.9)	38.9 (±9.9)	36.5 (±9.9)	34.0 (±9.1)
18–28	233 (44.3)	321 (34.4)	48 (17.9)	84 (26.4)	686 (33.5)
29–39	225 (42.8)	404 (43.2)	93 (34.7)	116 (36.5)	838 (41.0)
40–50	56 (10.6)	149 (16.0)	84 (31.3)	80 (25.2)	369 (18.0)
>50	12 (2.3)	60 (6.4)	43 (16.1)	38 (11.9)	153 (7.5)
**Sex**					
Male	394 (55.9)	60 (6.4)	160 (59.7)	258 (81.1)	772 (37.7)
Female	232 (44.1)	874 (93.6)	108 (40.3)	60 (18.9)	1274 (62.3)
**Education level**					
Master’s degree or above	526 (100)	3 (0.3)	-	19 (6.0)	548 (26.8)
Bachelor’s/Diploma	-	931 (99.7)	-	4 (1.3)	935 (45.7)
Higher secondary	-	-	13 (4.8)	107 (33.6)	120 (5.9)
Secondary or below	-	-	255 (95.1)	188 (59.1)	443 (21.6)
**Working experience** **(in years)**					
Mean (SD)	6.0 (±6.6)	9.4 (±8.5)	13.7 (±5.7)	12.9 (9.8)	9.6 (±8.8)
≤5	310 (64.4)	398 (45.5)	62 (24.6)	94 (31.40	864 (45.3)
6–10	86 (17.9)	195 (22.3)	37 (14.7)	51 (17.0)	369 (19.4)
≥11	85 (17.7)	281 (32.2)	153 (60.7)	154 (51.5)	673 (35.3)
**Working department**					
Medicine	223 (42.4)	316 (33.8)	63 (23.5)	28 (8.8)	630 (30.8)
Pediatric	90 (17.1)	142 (15.2)	37 (13.8)	7 (2.2)	276 (13.5)
Gyn–Obs	75 (14.3)	187 (20.0)	63 (23.5)	7 (2.2)	332 (16.2)
Surgery	138 (26.2)	289 (30.9)	83 (31.0)	20 (6.3)	530 (25.9)
Administration	-	-	22 (8.2)	256 (80.5)	278 (13.6)
**Hospital ownership**					
Public hospital	4520 (85.5)	844 (90.4)	217 (81.0)	284 (89.3)	1795 (87.7)
Private hospital	76 (14.5)	90 (9.6)	51 (19.0)	34 (10.7)	251 (12.3)

**Table 2 vaccines-11-00041-t002:** Uptake of hepatitis B vaccination among participating HCWs.

Variable	Physicians (N = 526)	Nurses (N = 934)	Cleaning Staff (N = 268)	Admin Staff (N = 318)	Total (N = 2046)
			n (%)		
**Received hepatitis B vaccine**	402 (76.4)	725 (77.6)	104 (38.8)	132 (41.5)	1363 (66.6)
**Tested blood (HBsAg) against hepatitis B virus antigen?**	357 (67.9)	749 (80.2)	95 (35.4)	128 (40.1)	1329 (65.0)
**HCWs who both tested blood and received the hepatitis B vaccine**	297 (56.5)	624 (66.8)	75 (28.0)	99 (31.1)	1095 (53.5)
**Willing to take the hepatitis B virus vaccine in the future for free**	N = 124	N = 209	N = 164	N = 186	(N = 683)
	110 (88.7)	193 (92.3)	145 (88.4)	166 (89.2)	614 (89.2)
**Had an occupational needlestick injury (NSI) in the past year**	114 (21.7)	380 (40.7)	53 (19.8)	12 (4.3)	559 (27.9)
**Number of times**					
Mean (SD)	2.1 (±0.8)	2.3 (±0.8)	2.4 (±0.7)	1.0 (±0.0)	2.2 (±0.8)
One time	36 (31.6)	82 (21.6)	8 (15.1)	12 (100)	138 (24.7)
Two times	33 (29.0)	105 (27.6)	14 (26.4)	-	152 (27.2)
≥ Three times	45 (39.5)	193 (50.8)	31 (58.5)	-	269 (48.1)
**Had NSI but unvaccinated**	22 (19.3)	81 (21.3)	34 (64.1)	10 (83.3)	147 (26.3)
**Attended training on hepatitis B in the last two years**	86 (16.3)	90 (9.6)	8 (2.9)	17 (5.4)	201 (9.8)
**Took extra protection during contact with hepatitis B patient**	420 (79.9)	817 (87.5)	59 (22.0)	74 (23.3)	1370 (67.0)
**Key recommended vaccines as essential for HCWs**					
Hepatitis B	358 (68.1)	680 (72.8)	111 (41.4)	126 (39.6)	1275 (62.3)
Influenza	230 (43.7)	370 (39.6)	43 (16.0)	67 (21.1)	710 (34.7)

**Table 3 vaccines-11-00041-t003:** Association between hepatitis B vaccination uptake and multiple predictors among vaccinated HCWs.

Variable	Vaccination Coverage (N = 1363)	Unadjusted Odds Ratio (UOR)	*p*-Value	Adjusted Odds Ratio (AOR)	*p*-Value
	n (%)	(95% CI)		(95% CI)	
**HCW type**					
Physicians	402 (29.5)	5.67 (4.02–7.98)	0.000	6.83 (4.72–9.89)	0.000
Nurses	725 (53.2)	5.62 (4.09–7.71)	0.000	6.02 (4.19–8.65)	0.000
Admin staff	132 (9.7)	0.97 (0.68–1.38)	0.881	1.06 (0.73–1.52)	0.769
Cleaning staff	104 (7.6)	Reference		Reference	
**Age (in years)**					
18–28	472 (34.6)	1.37 (0.93–2.02)	0.108	1.04 (0.60–1.81)	0.883
29–39	552 (40.5)	1.39 (0.96–2.02)	0.083	0.87 (0.55–1.40)	0.577
40–50	251 (18.4)	1.34 (0.89–2.00)	0.159	1.20 (0.77–1.86)	0.423
>50	88 (6.5)	Reference		Reference	
**Sex**					
Female	923 (67.7)	2.02 (1.65–2.48)	0.000	1.06 (0.81–1.40)	0.653
Male	440 (32.3)	Reference		Reference	
**Working experience** **(in years)**					
≤5	576 (42.3)	1.08 (0.86–1.37)	0.496	0.64 (0.43–0.95)	0.026
6–10	326 (23.9)	1.32 (1.00–1.78)	0.050	0.98 (0.69–1.41)	0.934
≥11	461 (33.8)	Reference		Reference	
**HCWs recommending hepatitis B vaccine**					
Yes	894 (65.6)	1.84 (1.49–2.28)	0.000	1.26 (1.01–1.61)	0.049
No	469 (34.4)	Reference		Reference	

## Data Availability

The authors are responsible for the data described in this manuscript. The datasets generated and analyzed are available from the corresponding author upon request.

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
