# Peer review of "Hepatitis B Vaccination Coverage among Bangladeshi Healthcare Workers: Findings from Tertiary Care Hospitals"

_vaccines, 2022, doi:10.3390/vaccines11010041_

Round 1
Reviewer 1 Report
Dear Editor, Dear Authors,
The manuscript entitled “Hepatitis B vaccination coverage among Bangladeshi healthcare workers: findings from tertiary care hospitals” addresses an important topic of hepatitis B vaccination among health care workers in LMICs and may provide a sound information to drive and improve vaccination policy.
There are some issues, however, which should be clarified in the content. Consider, please, the following:
*The key issue in the studies intended to provide descriptive data in some groups/populations is the representativeness of the sample. In the cross-sectional studies the key information is the response rate. Provide, please, what was the response rate / participation rate in Your study. Did participants differ to non-participants in according to some characteristics?
*Results provided in Table 1 suggest that the investigated sample of HCWs is not representative for Bangladesh (see Assessment of Healthcare Providers in Bangladesh 2021, page 23). Compare the descriptive demographic characteristics of the sample with the whole hired specific type workers (i.e. physicians, nurses …) available from administrative records. Discuss differences under Discussion.
*If some data (as mentioned on page 3 line 100-102) were collected from records therefore surely the data were not collected anonymously as mentioned on page 3 line 104.
*It would be useful to know, whether authors performed any sample size calculations to determine the number of participants in their study.
*The valuable information provided by the presented study is the hepatitis B vaccine coverage among different type HCWs. However, it would be useful to know what is the uncertainty level in the presented study. Provide, please, 95% CIs.
*It is not understandable why the admin staff group was used as a reference in the ORs calculations. It looks like it was a group which represents the requested standard, but it is not … Clarify, please.
*As described by authors, it was a comprehensive study, therefore the authors should provide more information about determinants of being non-vaccinated or of low vaccination level among cleaning staff (whether they were offered to have a vaccination? why they refused to take it? others …).
*It is not clear why the attendance of training on HBV was not considered in the logistic regression. In the descriptive part it would be worth knowing the proportion of those who were offered training, and next how many of those who were offered finally participated in.
*It would be useful to know also whether the experience of the exposure (NSI, other) influenced vaccination status.
*Authors argue for national policy, but it is not know what determinants should be targeted by such policy to get better vaccination response.
*The result mentioned under Discussion, page 8 line 244-245 is not statistically significant, therefore should NOT be used for any conclusions. The same for lines 252-256.
*page 2, line 53 “three 1 doses” … typing error?
Regards,
AG, Reviewer
Author Response
Dear Reviewer,
Greetings! Thank you very much for your detailed review and very useful comments on our manuscript "Hepatitis B vaccination coverage among Bangladeshi healthcare workers: findings from tertiary care hospitals." We appreciate your suggestions and insightful comments to enrich our manuscript.
*The key issue in the studies intended to provide descriptive data in some groups/populations is the representativeness of the sample. In the cross-sectional studies the key information is the response rate. Provide, please, what was the response rate / participation rate in Your study. Did participants differ to non-participants in according to some characteristics?
Response: Thank you for bringing to light a key issue. We have added the non-response rate ( it was 4%). in the revised manuscript. We have approached 2131 HCWs; among them, 2046 participated (in line 149).
*Results provided in Table 1 suggest that the investigated sample of HCWs is not representative for Bangladesh (see Assessment of Healthcare Providers in Bangladesh 2021, page 23). Compare the descriptive demographic characteristics of the sample with the whole hired specific type workers (i.e. physicians, nurses …) available from administrative records. Discuss differences under Discussion.
Response: Thank you for the suggestion. We have looked into the Report, compared our demographic data and provided a justification of the difference in the Discussion section in lines 84-89
*If some data (as mentioned on page 3 line 100-102) were collected from records therefore surely the data were not collected anonymously as mentioned on page 3 line 104.
Response: Response: Thank you for your feedback. The data on vaccination status, blood test history against the hepatitis B virus, or the number of needlestick injuries were all collected from the HCW’s recall, and at times if they could not remember their vaccination status, they were asked to confirm by checking their vaccine card, termed as “records’ in line 102. Therefore, the data collectors did not have direct access to any record. The information was collected from the participant. To avoid confusion, we have removed the word “record” from line 119.
*It would be useful to know, whether authors performed any sample size calculations to determine the number of participants in their study.
Response: No, we collected data from approximately 25% of HCWs from each category at each study site to ensure proper representatives of each healthcare facility’s workforce, line 105.
*The valuable information provided by the presented study is the hepatitis B vaccine coverage among different type HCWs. However, it would be useful to know what is the uncertainty level in the presented study. Provide, please, 95% CIs.
Response: We have calculated the 95% CI and added it in the revised manuscript in line 163.
*It is not understandable why the admin staff group was used as a reference in the ORs calculations. It looks like it was a group which represents the requested standard, but it is not … Clarify, please.
Response: Thank you for the feedback. Considering the healthcare workers who were directly involved in patient service, we consider cleaning staff instead of administrative staff as a reference and reanalyze data for Table 3, page 7.
*As described by authors, it was a comprehensive study, therefore the authors should provide more information about determinants of being non-vaccinated or of low vaccination level among cleaning staff (whether they were offered to have a vaccination? why they refused to take it? others …).
Response: Thank you for raising the issue. However, we didn’t explore any determinants of low vaccine uptake in this study and revised them in discussion line 202.
*It is not clear why the attendance of training on HBV was not considered in the logistic regression. In the descriptive part it would be worth knowing the proportion of those who were offered training, and next how many of those who were offered finally participated in.
Response: Thank you for pointing out this critical issue. We exclude the training on HBV from regression due to a higher p-value (0.35) from the univariate model. We only considered Variables with a p-value of 0.25 from the univariate model and mentioned in section 2.4 Statistical Analyses, line no 131. We did not collect that information on whether the hospital offered the HBV training or not.
*It would be useful to know also whether the experience of the exposure (NSI, other) influenced vaccination status.
Response: Thank you for the suggestion. However, ‘exposure of NSI’ was excluded due to a higher p-value (> 0.25) from the univariate model.
*Authors argue for national policy, but it is not known what determinants should be targeted by such policy to get better vaccination response.
Response: Thank you. We have added ‘of mandatory vaccination’ in the discussion section line 212.
*The result mentioned under Discussion, page 8 line 244-245 is not statistically significant, therefore should NOT be used for any conclusions. The same for lines 252-256.
Response: Thank you for raising this concern. In the revised regression model, after changing the reference category of HCW, including the revised age group (suggested by another reviewer), we also modified the writing in the discussion section line 270-273. Besides, in line 274, we have added ‘from the univariate model’ at the beginning of the statement. We want to keep this association (as it is significant now) along with references.
*page 2, line 53 “three 1 doses” … typing error?
Response: Yes, it was a typing error and has been removed. Thank you for the observation.
Kind regards,
Md. Golam Dostogir Harun
Review response submission
Date: 19 December 2022

Reviewer 2 Report
1. These investigators undertook a survey to determine hepatitis B vaccination uptake in healthcare workers in Bangladesh tertiary care hospitals. This cross-sectional study involved 11 hospitals. 2046 healthcare workers participated in this study. The information was collected through face-to-face interviews in the local language. Approximately 25% of healthcare workers in each category were interviewed in each study hospital. The categories included physicians, nurses, cleaning staff, and administrative staff.
2. Overall, 66.6% of the healthcare workers had received hepatitis B vaccine. The percentage was higher in physicians and nurses and lower in cleaning staff and administrative staff. The study is a well-done study. The analysis identified factors associated with vaccine uptake. It is identified potential strategies which might increase vaccine uptake in all healthcare workers in these hospitals. The authors note potential limitations which include bias in recall of vaccination status and in recall of the number of needlestick injuries.
Author Response
Dear Reviewer,
Greetings! Thank you very much for your positive remarks and valuable comments on our manuscript "Hepatitis B vaccination coverage among Bangladeshi healthcare workers: findings from tertiary care hospitals." We appreciate your suggestions and insightful comments to enrich our manuscript.
- These investigators undertook a survey to determine hepatitis B vaccination uptake in healthcare workers in Bangladesh tertiary care hospitals. This cross-sectional study involved 11 hospitals. 2046 healthcare workers participated in this study. The information was collected through face-to-face interviews in the local language. Approximately 25% of healthcare workers in each category were interviewed in each study hospital. The categories included physicians, nurses, cleaning staff, and administrative staff.
- Overall, 66.6% of the healthcare workers had received hepatitis B vaccine. The percentage was higher in physicians and nurses and lower in cleaning staff and administrative staff. The study is a well-done study. The analysis identified factors associated with vaccine uptake. It is identified potential strategies which might increase vaccine uptake in all healthcare workers in these hospitals. The authors note potential limitations which include bias in recall of vaccination status and in recall of the number of needlestick injuries.
Kind regards,
Md. Golam Dostogir Harun
Review response submission
Date: 19 December 2022
Reviewer 3 Report
The study presented by Harun et al is a study describing hepatitis B vaccination coverage in HCWs from Bangladeshi hospitals. In addition, the authors also showed which factors may be related to the increase in this coverage. Despite being an interesting study, which sheds light on this issue in this population, some points need to be better clarified to make the study even more cohesive.
1) Major:
Introduction:
- Line 37: "It is a major public health threat that affects 296 million people around the world" - This sentence could be rephrased, the way it is presented suggests this number refers to new annualHepatitis B cases. However, according to the WHO, this number refers to the sum of people living with chronic infection until the year 2019.
- Lines 45-46: "HCWs are therefore classified as a high priority and are recommended to take the hepatitis B vaccine as a standard precaution" - Where? All over the world? In Bangladesh? Please Make it clearer.
- Line 69-70: "to determine the prevalence of hepatitis B vaccine uptake" - The word vaccine prevalence does not seem appropriate in this context, since the study did not assess the actual prevalence of anti-HBs in this population. Perhaps it would be better to use the term frequency of vaccination, for example.
Materials and Methods:
- Lines 77-81- "2.1. Study Design and Settings''- Could the authors further explain how the selection criteria for the 11 hospitals studied were established? Please provide a written explanation in the text of the article.
- Lines 84-85 - "including physicians, nurses, and 84 cleaning staff from the departments ofMedicine, Pediatrics, Surgery, and Gyn-Obs"- Explain better what is done in the Medicine andGyn-Obs departments.
- Line 86 - "DGHS"- Explain the meaning of this term.
- Line 91 -"Approximately 25% of HCWS"- 25% of the total population? 25% of hospitals?Explain further. And why only 25%?
- Line 113 - "age group (18-25, 26-45, and ≥46 years)" - Why were the age groups divided in this way? It would be more homogeneous if the authors divided the age groups into ±10 years (e.g.18-28; 29-39; 40-46 and >46), which is more or less the general average of the population studied (±9.1 years old).
- Lines 144-146 - "More than three-quarters of physicians and nurses obtained the hepatitis B Vaccine, but only 41.1% of administrative staff and 38.8% of cleaning staff were found to be vaccinated against hepatitis B" - How many doses did these individuals have? It would be very important and interesting to have these dose percentages.
- Lines 146-148 - "While nearly two-thirds (65.0%) of participants had a blood test for the vaccine antibody, only 53.5% of HCWs had performed both a blood test and hepatitis B vaccine uptake" -The sentence is not very clear. What type of test for HBV? Test for assessment of immunity or serum conversion (anti-HBs), assessment of present (HBsAg) or past (anti-HBc IgG) infection? The Sentence needs to be improved and the questions need to be better explained.
Discussion:
- Lines 183 and 213 - "LMICs" - Explain the meaning of this term.
2) Minor:
- Authors should be aware of the journal's reference format. The model used in this document does not follow the recommended standards.
Author Response
Dear Reviewer,
Greetings! Thank you very much for your detailed review and very useful comments on our manuscript "Hepatitis B vaccination coverage among Bangladeshi healthcare workers: findings from tertiary care hospitals." We appreciate your suggestions and insightful comments to enrich our manuscript.
Comments and Suggestions for Authors
The study presented by Harun et al is a study describing hepatitis B vaccination coverage in HCWs from Bangladeshi hospitals. In addition, the authors also showed which factors may be related to the increase in this coverage. Despite being an interesting study, which sheds light on this issue in this population, some points need to be better clarified to make the study even more cohesive.
1) Major:
Introduction:
- Line 37: "It is a major public health threat that affects 296 million people around the world" - This sentence could be rephrased, the way it is presented suggests this number refers to new annual Hepatitis B cases. However, according to the WHO, this number refers to the sum of people living with chronic infection until the year 2019.
Response: Revision has been made as advised. The revised line stands “It is a major public health threat that has affected 296 million people around the world until the year 2019” in line 39.
- Lines 45-46: "HCWs are therefore classified as a high priority and are recommended to take the hepatitis B vaccine as a standard precaution" - Where? All over the world? In Bangladesh? Please Make it clearer.
Response: It is recommended for HCWs all over the world by CDC, rephrased the sentence in lines 46-48. Thanks for feedback.
- Line 69-70: "to determine the prevalence of hepatitis B vaccine uptake" - The word vaccine prevalence does not seem appropriate in this context, since the study did not assess the actual prevalence of anti-HBs in this population. Perhaps it would be better to use the term frequency of vaccination, for example.
Response: This is an excellent suggestion. We have reconsidered and changed the sentence to “to determine the coverage of hepatitis B vaccination” in line 74.
Materials and Methods:
- Lines 77-81- "2.1. Study Design and Settings''- Could the authors further explain how the selection criteria for the 11 hospitals studied were established? Please provide a written explanation in the text of the article.
Response: Thank you for the very vital feedback. We have included a written justification for including the 11 hospitals in lines 84-88 of the revised manuscript, “These 11 hospitals were purposively selected by the government authorities (DGHS) and, represent a quarter of all the country's tertiary level hospitals, and a third of the public facilities of this scale.”
- Lines 84-85 - "including physicians, nurses, and 84 cleaning staff from the departments of Medicine, Pediatrics, Surgery, and Gyn-Obs"- Explain better what is done in the Medicine and Gyn-Obs departments.
Response: Thank you. We have written details in the revised manuscript.
- Line 86 - "DGHS"- Explain the meaning of this term.
Response: DGHS stands for the Directorate General of Health Services. It is the division under the Ministry of Health and Family Welfare responsible for health services in Bangladesh. Added to the manuscript as well.
- Line 91 -"Approximately 25% of HCWS"- 25% of the total population? 25% of hospitals? Explain further. And why only 25%?
Response: Thank you. We have enrolled approximately 25% of HCWs randomly from each study site to ensure proper representatives of the study participants, line 105.
- Line 113 - "age group (18-25, 26-45, and ≥46 years)" - Why were the age groups divided in this way? It would be more homogeneous if the authors divided the age groups into ±10 years (e.g.18-28; 29-39; 40-46 and >46), which is more or less the general average of the population studied (±9.1 years old).
Response: Thank you. We have revised the age category (18-28, 29-39, 40-50 and >50 years) for analysis in table 1, table 3, and in the section in statistical analyses, line 132.
- Lines 144-146 - "More than three-quarters of physicians and nurses obtained the hepatitis B Vaccine, but only 41.1% of administrative staff and 38.8% of cleaning staff were found to be vaccinated against hepatitis B" - How many doses did these individuals have? It would be very important and interesting to have these dose percentages.
Response: Thank you for your great observation. We have considered vaccinated HCWs who completed all three doses of the Hepatitis B vaccine. It has been included in the revised manuscript in the methods section, line 116.
- Lines 146-148 - "While nearly two-thirds (65.0%) of participants had a blood test for the vaccine antibody, only 53.5% of HCWs had performed both a blood test and hepatitis B vaccine uptake" -The sentence is not very clear. What type of test for HBV? Test for assessment of immunity or serum conversion (anti-HBs), assessment of present (HBsAg) or past (anti-HBc IgG) infection? The Sentence needs to be improved and the questions need to be better explained.
Response: We appreciated the reviewer’s suggestion. We have added the test name in the data collection section line 110-111 (of HBsAg to assess the presence of), revised in lines 164-165 (HBsAg blood tests to determine the presence of hepatitis) and Table 2
Discussion:
- Lines 183 and 213 - "LMICs" - Explain the meaning of this term.
Response: Apologies for not expanding the abbreviation upon first use. Revised.
2) Minor:
- Authors should be aware of the journal's reference format. The model used in this document does not follow the recommended standards.
Response: Thank you for your suggestion. We have updated the reference to the AMA format.
Kind regards,
Md. Golam Dostogir Harun
Review response submission
Date: 19 December 2022
Reviewer 4 Report
This paper is straightforward. I think it is of average to fair interest but of some importance. It would be nice if it had some policy suggestions in the manuscript in the conclusions, but other than that, the study is well-done and well-edited. It is nice to review a manuscript so clear and well-edited.
I would comment that the authors switch to first person occasionally in the manuscript, which is distracting.
Author Response
Dear Reviewer,
Greetings! Thank you very much for your detailed review and very useful comments on our manuscript "Hepatitis B vaccination coverage among Bangladeshi healthcare workers: findings from tertiary care hospitals." We appreciate your suggestions and insightful comments to enrich our manuscript.
This paper is straightforward. I think it is of average to fair interest but of some importance. It would be nice if it had some policy suggestions in the manuscript in the conclusions, but other than that, the study is well-done and well-edited. It is nice to review a manuscript so clear and well-edited.
I would comment that the authors switch to first person occasionally in the manuscript, which is distracting.
Response: Thank you so much for your positive remarks. We have included a statement related to policy implications in the recommendation and conclusion section of the revised manuscript.
Kind regards,
Md. Golam Dostogir Harun
Review response submission
Date: 19 December 2022
Reviewer 5 Report
This research and manuscript focus on obtaining insights into Hepatitis B vaccination coverage among different types of healthcare workers at 11 tertiary care hospitals in Bangladesh. Face-to-face interviews were conducted with 2,046 healthcare workers between September 2020 and January 2021.
· More information needs to be provided regarding why tertiary care hospitals were the focus of the study and why these 11 tertiary care hospitals were selected. If possible, it would be helpful to provide more information regarding why Hepatitis B is a health threat in these facilities, including which types of healthcare workers are most at risk for infection.
· It would help to provide more information regarding Hepatitis B vaccination recommendations for healthcare workers in Bangladesh (such as, are there recommendations) and healthcare worker access to Hepatitis B vaccine (e.g., is it widely available? is there a cost? Where do healthcare workers go if they want to receive Hepatitis B vaccination?)
· It should be made clear, including in the title, this is an exploratory study, including because it involves purposely selected healthcare facilities and the use of convenience sampling. Also, the manuscript should consistently state the focus of the study was on learning more about Hepatitis B infection and vaccination in these 11 tertiary care hospitals (versus tertiary care hospitals in Bangladesh more broadly). The manuscript does do this in Lines 70-72 and this should be done throughout.
· Lines 61-63 – it would help to indicate how the identified factors are related to Hepatitis B vaccination, particularly those that are ambiguous, such as “hospital infection control situation” and “type of HCW.”
· This study involves convenience sampling and as such, more details and specifics need to be provided regarding how potential participants were identified, recruited, and selected at each of the 11 tertiary care centers. It would also be helpful to provide information regarding the participation rate; that is, what percentage of those approached participated in the survey. It is stated that “all HCW groups and administrative staff from the department of Medicine, Pediatrics, Surgery, and Gyn-Obs” were included but no information is provided regarding how many individuals total and by each group were potential respondents.
· It is mentioned that the study used “a semi-structured questionnaire,” but no information is provided regarding open-end questions or how responses to open-end questions were analyzed. Based on the information provided in Section 2.3, it is not clear what made this a “semi-structured” questionnaire.
· It is stated that some information was provided based on “respondent’s recall and records,” but no information is provided regarding what records were used and how discrepancies were handled.
· Lines 110-111 – it is not clear what is meant by “the cluster effect of different hospitals.”
· The information in Table 2 regarding occupational needlestick injury in the past year indicates that 114 physicians reported a needlestick but also indicates that 220 had an NSI and were unvaccinated. Those numbers should be checked.
· It would be helpful in the Discussion to discuss the process and potential barriers and facilitators of the recommended changes in Bangladesh. For example, what would it take for there to be 1) a national recommendation or policy requiring HCWs to receive Hepatitis B vaccination, and who would make it, 2) more training regarding Hepatitis B disease and vaccine in tertiary care and/or more broadly, 3) free or lower cost Hepatitis B vaccine for HCWs, and 4) Hepatitis B vaccination campaigns directed at HCWs?
· The Discussion would also benefit from recommendations regarding needed future research in Bangladesh with respect to Hepatitis B infection and vaccination in hospitals and among healthcare workers.
· The Limitation section (lines 263-266) lacks depth and adequate description of how these limitations matter. In addition, a major limitation – the use of a convenience sample – needs to be included and discussed.
· There are many places were the wording could be strengthened, including:
o Lines 20-21: better wording would be “Descriptive and multivariate STATISTICS WERE USED TO ANALYZE THE DATA.”
o Line 30: better wording would be “Low uptake of Hepatitis B vaccination among HCWs SUGGESTS POLICIES THAT REQUIRE VACCINATION ARE NEEDED TO ACHIEVE OPTIMUM VACCINE COVERAGE.”
o Line 53 – there is an unneeded character at the end of the line
o Lines 86 – better wording would be “Before initiating the study were OBTAINED written permission from DGHS. . .”
o Lines 155-156 – better wording would be “However, 62.3% of respondents agreed that Hepatitis B vaccine was essential and should be mandatory in our country.”
o Lines 217-221 – the wording here is confusing
o Lines 268-273 – the Conclusion is also poorly worded
Author Response
Dear Reviewer,
Greetings! Thank you very much for your detailed review and very useful comments on our manuscript "Hepatitis B vaccination coverage among Bangladeshi healthcare workers: findings from tertiary care hospitals." We appreciate your suggestions and insightful comments to enrich our manuscript.
This research and manuscript focus on obtaining insights into Hepatitis B vaccination coverage among different types of healthcare workers at 11 tertiary care hospitals in Bangladesh. Face-to-face interviews were conducted with 2,046 healthcare workers between September 2020 and January 2021.
More information needs to be provided regarding why tertiary care hospitals were the focus of the study and why these 11 tertiary care hospitals were selected. If possible, it would be helpful to provide more information regarding why Hepatitis B is a health threat in these facilities, including which types of healthcare workers are most at risk for infection.
Response: We have taken your feedback into consideration and revised the "Study Design and Settings" section accordingly. "These 11 hospitals were purposively selected by the government authorities (the Ministry of Health) and, represent a quarter of all the country's tertiary level hospitals and a third of the public facilities of this scale. Tertiary, the highest level, hospitals or institutes have medical colleges and universities, specialized departments, and are usually referral hospitals, where patients from all over the country seek medical help. The average bed capacity and annual patient turnover of selected hospitals ranged from 450-2600 and 15,000-85,000, respectively. The high patient volume makes the transmission of infectious diseases a serious concern in these facilities." lines 84-88.
It would help to provide more information regarding Hepatitis B vaccination recommendations for healthcare workers in Bangladesh (such as, are there recommendations) and healthcare worker access to Hepatitis B vaccine (e.g., is it widely available? is there a cost? Where do healthcare workers go if they want to receive Hepatitis B vaccination?)
Response: This is very useful feedback for justifying the purpose of our study. Even though the Centers for Disease Control and Prevention (CDC) deemed HCWs as a population high-risk population and recommended vaccination against HBV, no recommendation or policy for HCWs exists in Bangladesh. We have discussed this in detail in the manuscript in lines 61 to 66.
It should be made clear, including in the title, this is an exploratory study, including because it involves purposely selected healthcare facilities and the use of convenience sampling. Also, the manuscript should consistently state the focus of the study was on learning more about Hepatitis B infection and vaccination in these 11 tertiary care hospitals (versus tertiary care hospitals in Bangladesh more broadly). The manuscript does do this in Lines 70-72 and this should be done throughout.
Response: Response: Thank you for your suggestion. We have added “tertiary care hospital” in lines 20 and 178. Our investigated hospitals represent a quarter of all tertiary care hospitals in Bangladesh and represent the overall situation of all tertiary care hospitals. Therefore we considered keeping the title unchanged.
Lines 61-63 – it would help to indicate how the identified factors are related to Hepatitis B vaccination, particularly those that are ambiguous, such as “hospital infection control situation” and “type of HCW.”
Response: The line has been revised. Thank you.
This study involves convenience sampling and as such, more details and specifics need to be provided regarding how potential participants were identified, recruited, and selected at each of the 11 tertiary care centers. It would also be helpful to provide information regarding the participation rate; that is, what percentage of those approached participated in the survey. It is stated that “all HCW groups and administrative staff from the department of Medicine, Pediatrics, Surgery, and Gyn-Obs” were included but no information is provided regarding how many individuals total and by each group were potential respondents.
Response: Thank you for the comment. We have included the overall participation response in section 3.1, line 146 and revised the participants' statement in lines 94-96
It is mentioned that the study used “a semi-structured questionnaire,” but no information is provided regarding open-end questions or how responses to open-end questions were analyzed. Based on the information provided in Section 2.3, it is not clear what made this a “semi-structured” questionnaire.
Response: We have added in the analyses section line 127-128. “Response for the open-ended question of “Number of needlestick injuries” and “Recommended vaccines that essential for HCWs” were identified and categorized.”
It is stated that some information was provided based on “respondent’s recall and records,” but no information is provided regarding what records were used and how discrepancies were handled.
Response: Thank you for your feedback. The data on vaccination status, blood test history against the hepatitis B virus, or the number of needlestick injuries were all collected from the HCW’s recall, and at times if they could not remember their vaccination status they were asked to confirm by checking their vaccine card, termed as “records’ in line 119. Therefore, the data collectors did not have direct access to any record. To avoid confusion, we have removed the word “record” from line 119.
Lines 110-111 – it is not clear what is meant by “the cluster effect of different hospitals.”
Response: This study included 11 tertiary hospitals and there may be a potential for a correlation of outcomes between these hospitals that can alter the association. To minimize this clustering effect, we have performed mixed effects logistic regression analysis.
The information in Table 2 regarding occupational needlestick injury in the past year indicates that 114 physicians reported a needlestick but also indicates that 220 had an NSI and were unvaccinated. Those numbers should be checked.
Response: We want to apologize for the typing error. The data is supposed to be “22”, thank you for identifying the mistake.
It would be helpful in the Discussion to discuss the process and potential barriers and facilitators of the recommended changes in Bangladesh. For example, what would it take for there to be 1) a national recommendation or policy requiring HCWs to receive Hepatitis B vaccination, and who would make it, 2) more training regarding Hepatitis B disease and vaccine in tertiary care and/or more broadly, 3) free or lower cost Hepatitis B vaccine for HCWs, and 4) Hepatitis B vaccination campaigns directed at HCWs?
Response: Thank you very much for your kind remarks and very important suggestions. We have incorporated policy recordation in the revised manuscript (in lines 280-284)
The Discussion would also benefit from recommendations regarding needed future research in Bangladesh with respect to Hepatitis B infection and vaccination in hospitals and among healthcare workers.
Response: We have added to the revised Discussion. Thank you
The Limitation section (lines 263-266) lacks depth and adequate description of how these limitations matter. In addition, a major limitation – the use of a convenience sample – needs to be included and discussed.
Response: Thank you. We have added the limitation more clearly. The study randomly enrolled 25% of participants from each study site to ensure the representativeness of all categories of HCWS..
There are many places were the wording could be strengthened, including:
o Lines 20-21: better wording would be “descriptive and multivariate statistics were used to analyze the data.”
Response: Thank you, we have revised.
o Line 30: better wording would be “Low uptake of Hepatitis B vaccination among HCWs SUGGESTS POLICIES THAT REQUIRE VACCINATION ARE NEEDED TO ACHIEVE OPTIMUM VACCINE COVERAGE.”
Response: Thank you, we have revised.
o Line 53 – there is an unneeded character at the end of the line
Response: Thank you, we have revised.
o Lines 86 – better wording would be “Before initiating the study were OBTAINED written permission from DGHS. . .”
Response: Thank you, we have revised. Accordingly.
o Lines 155-156 – better wording would be “However, 62.3% of respondents agreed that Hepatitis B vaccine was essential and should be mandatory in our country.”
Response: Thank you, we have revised.
o Lines 217-221 – the wording here is confusing
Response: Thank you, we have corrected the sentences for better clarification.
o Lines 268-273 – the Conclusion is also poorly worded
Response: Thank you for your keen observation and feedback. We have revised the manuscript and updated the languages appropriately. The conclusion looks better now.
Kind regards,
Md. Golam Dostogir Harun
Review response submission
Date: 19 December 2022
Round 2
Reviewer 1 Report
Dear Editor,
Dear Authors,
The corrections and explanations provided clarify the content of the manuscript and study design. In my opinion the current form of the paper is suitable for publication.
Congratulations,
AG, Reviewer
Author Response
Dear Reviewer,
Greetings! Thank you so much again for your kind review and valuable comments on our manuscript "Hepatitis B vaccination coverage among Bangladeshi healthcare workers: findings from tertiary care hospitals." We highly appreciate your suggestions and insightful comments to enrich our manuscript.
Kind regards,
Md. Golam Dostogir Harun
Review response submission
Date: 22 December 2022
Reviewer 3 Report
The requested corrections have been answered. However, two points need further explanation:
in section "2.3. data collection":
Lines 107-108 - "of HBsAg to assess the presence of hepatitis B virus" - It is still not clear whether this serological test was done during the study after blood collection, or whether the respondent was asked if he/she had ever previously had this test done in his/her life.
in section "Results":
table 2, row 2 - "Tested blood (HBsAg) against hepatitis B Virus antibody?" Wouldn't it be antigen instead of antibody? as shown in the methodology section HBsAg antigen was evaluated.
Author Response
Dear Reviewer,
Greetings! Thank you so much again for your review and valuable comments on our manuscript "Hepatitis B vaccination coverage among Bangladeshi healthcare workers: findings from tertiary care hospitals." We highly appreciate your suggestions and insightful comments to enrich our manuscript.
in section "2.3. data collection":
Lines 107-108 - "of HBsAg to assess the presence of hepatitis B virus" - It is still not clear whether this serological test was done during the study after blood collection, or whether the respondent was asked if he/she had ever previously had this test done in his/her life.
Response: Thank you so much for the remark. The respondent was asked if he/she had ever previously had this test done in his/her life. We have modified it clearly in the revised manuscript ( page-3, lines 107-108).
in section "Results":
table 2, row 2 - "Tested blood (HBsAg) against hepatitis B Virus antibody?" Wouldn't it be antigen instead of antibody? as shown in the methodology section HBsAg antigen was evaluated.
Response: Thank you so much for the observation. You are right; this will be “Tested blood (HBsAg) against hepatitis B Virus antigen.” We have corrected this in the revised manuscript (in table 2, row 2).
Kind regards,
Md. Golam Dostogir Harun
Review response submission
Date: 22 December 2022